# Impact of cilostazol on clinical outcomes in lower extremity arterial disease patients after angioplasty: A real-world analysis

Hsien-Yuan Chang[1,2], Hui-Wen Lin[2], Po-Wei Chen[1,2], Sheng-Hsiang Lin[1,3,4], Ting-Hsing Chao[2,5☯*], Yi-Heng Li[2]☯*

1 Institute of Clinical Medicine, College of Medicine, National Cheng Kung University, Tainan, Taiwan, 2 Division of Cardiology, Department of Internal Medicine, National Cheng Kung University Hospital, College of Medicine, National Cheng Kung University, Tainan, Taiwan, 3 Biostatistics Consulting Center, National Cheng Kung University Hospital, College of Medicine, National Cheng Kung University, Tainan, Taiwan, 4 Department of Public Health, College of Medicine, National Cheng Kung University, Tainan, Taiwan, 5 Division of Cardiology, Department of Internal Medicine, Chung-Shan Medical University Hospital, School of Medicine, Chung Shan Medical University, Taichung, Taiwan

☯ These authors contributed equally to this work.
* heng@mail.ncku.edu.tw (YHL); chaotinghsing@gmail.com (THC)

## Abstract

### Background

Cilostazol has been shown to improve walking distance in patients with lower extremity arterial disease (LEAD) and may reduce restenosis after revascularization. However, its long-term prognostic impact in real-world settings remains underexplored.

### Methods

We conducted a retrospective cohort study using data from Taiwan's National Health Insurance Research Database (2012–2022). We included stable LEAD patients who had undergone percutaneous transluminal angioplasty (PTA) and remained event-free for 1 year. Stabilized inverse probability of treatment weighting (IPTW) was applied to adjust for baseline confounders. The study aimed to evaluate the effect of cilostazol on major adverse cardiovascular events (MACE), major adverse limb events (MALE), and composite bleeding outcomes.

### Results

Among 5,300 stable LEAD patients, of whom 844 received cilostazol alone, 1,786 received aspirin or clopidogrel, and 2,670 received cilostazol combined with aspirin or clopidogrel. After IPTW, there were no significant differences between cilostazol monotherapy and any antiplatelet therapy groups regarding MACE, MALE, or composite bleeding outcomes (aHR [95% CI] = 0.84 [0.68–1.03], p = 0.09; 0.84 [0.70–1.01], p = 0.06; 0.88 [0.71–1.10], p = 0.26, respectively). In secondary outcomes,

**Data availability statement:** The minimal data set required to replicate the findings has been provided via this manuscript's Supporting Information files. The original data derived from the National Health Insurance Research Database (NHIRD), Taiwan, are not publicly available due to legal and ethical restrictions. Access to these data requires formal application and approval by the Health and Welfare Data Science Center, Ministry of Health and Welfare, Taiwan. Requests for data access can be directed to the Health and Welfare Data Science Center via email (e.g., Mr. Wu at stwu@mohw. gov.tw) or through the official website: https:// dep.mohw.gov.tw/DOS/np-2497-113.html. The website provides detailed information on data application procedures and official contact points.

**Funding:** This work was supported by a grant from the Chung Shan Medical University Hospital (CSH-2025-D-002, Ting-Hsing Chao).

cilostazol treatment was associated with a reduced rate of subsequent angioplasty (aHR [95% CI] = 0.80 [0.60–0.98], p = 0.03). There were no significant differences in clinical outcomes when comparing cilostazol monotherapy to cilostazol combined with antiplatelet therapy.

## Conclusion

In this real-world Asian cohort, cilostazol showed similar prognostic benefits and safety compared to standard antiplatelet therapy. These findings support its role in the long-term management of LEAD patients following PTA, particularly in Asian populations.

## Introduction

Lower extremity arterial disease (LEAD) ranks as the third leading cause of death among individuals with atherosclerotic cardiovascular diseases, following coronary artery disease and stroke. In recent years, advancements in endovascular revascularization have positioned it as a cornerstone in the management of symptomatic LEAD, especially for patients with intermittent claudication or critical limb ischemia. However, despite the initial technical success of these interventions, high rates of restenosis and recurrent symptoms remain prevalent. [1] These challenges underscore the importance of implementing guideline-directed medical therapy to optimize long-term clinical outcomes and maintain vascular patency following revascularization [2–4]

Cilostazol, a phosphodiesterase 3 inhibitor with antiplatelet and vasodilatory properties, is commonly used in LEAD patients to alleviate the symptoms of intermittent claudication and to reduce post-angioplasty restenosis. Several recent randomized controlled trials have demonstrated its efficacy in lowering late lumen loss and reducing the need for target lesion revascularization. [5] Beyond its vascular benefits, cilostazol has shown promising effects on angiogenesis in preclinical studies, [6–8] and emerging evidence suggests its potential in reducing major adverse cardiovascular events (MACE) [9,10]. However, despite these findings, current clinical guidelines do not recommend cilostazol for the explicit purpose of MACE reduction, [2] and the European Society of Cardiology guidelines make no recommendation of cilostazol at all in the context of LEAD management. [4] Notably, although randomized controlled trials have highlighted the potential benefits of cilostazol, evidence from real-world clinical practice—particularly among LEAD patients following angioplasty—remains relatively limited. This discrepancy between trial-based findings and current clinical guideline recommendations invites further investigation into cilostazol's role in the management of LEAD in routine care settings.

Therefore, this study aims to compare cilostazol with other current antiplatelet therapies using real-world data, evaluating their impact on clinical outcomes, including MACE and major adverse limb events (MALE), in patients with LEAD undergoing endovascular revascularization in a real-world setting.

## Methods

### Study design

This is a retrospective cohort study using data from the National Health Insurance Research Database in Taiwan, which were accessed and processed for research purposes between November 1, 2024, and March 31, 2025, covering the period from January 1, 2012, to December 31, 2022. This study adhered to the Declaration of Helsinki and received approval from the Institutional Review Board of the National Cheng Kung University Hospital (IRB number: A-EX-113–007). The requirement for informed consent was waived owing to the retrospective design of the study. All data were fully anonymized before access, and the authors did not have access to any information that could identify individual participants during or after data collection

### Study population and inclusion criteria

We enrolled adult patients who were newly diagnosed with LEAD and underwent percutaneous transluminal angioplasty (PTA). Medical information was recorded using the International Classification of Diseases, Ninth and Tenth Revisions, Clinical Modification (ICD-9-CM and ICD-10-CM) codes. Patients diagnosed with LEAD were identified using the following codes: ICD-9-CM codes (443.9, 440.2, 440.3, 440.4) and ICD-10-CM codes (I65.9, I63.00, I63.10, I63.20, I63.29, I73.9, I70.2, I70.3, I70.4, I70.5, I70.6, I70.7, I70.8). Patients who underwent PTA were identified using the following codes: ICD-9-CM codes (38.08, 38.18, 38.38, 38.48, 38.58, 35.68, 38.88, 39.50, 39.7, 39.90, 39.25, 39.26, 39.29) and ICD-10-CM codes (041, 045, 047, 049, 04B, 04C, 04H, 04J, 04L, 04N, 04P, 04Q, 04R, 04S, 04U, 04V, 04W).

In line with current guidelines that recommend dual antiplatelet therapy (DAPT) for 1–3 months post-PTA, [2,4] To avoid immortal time bias, a landmark design was employed, [11] with the index date set between six months and one year after PTA. Clinical information collected during the index date included age, sex, comorbidities, and medications. Detailed corresponding ICD codes are listed in Supplemental Table 1. Patients were grouped according to the medications they were using at the index date. To focus the analysis on the long-term effects of the medications on clinical outcomes, patients who experienced major adverse cardiovascular events (MACE) within the first year following PTA or major adverse limb events (MALE) between six months and one year post-angioplasty were excluded from the study (Supplemental Fig 1). Additional exclusion criteria included medication records of fewer than 84 days within the index date, use of aspirin plus clopidogrel, previous history of atrial fibrillation, or other indications for novel oral anticoagulants or warfarin.

### Clinical outcomes

Clinical outcomes were recorded starting from one year after PTA. Detailed corresponding ICD codes are listed in Supplemental Table 1. The primary endpoint was the incidence of MACE, defined as a composite outcome of cardiovascular death, non-fatal myocardial infarction, stroke, or transient ischemic attack. The co-primary endpoint was the incidence of MALE, defined as a composite outcome of undergoing PTA for LEAD or amputation. The secondary endpoints were cardiovascular death, non-fatal myocardial infarction, stroke or transient ischemic attack, undergoing PTA for LEAD, and amputation. The safety outcome was a composite of bleeding events, including hemorrhagic stroke, gastrointestinal bleeding, or other site bleeding.

### Statistical analysis

Continuous data are presented as mean ± standard deviation, while categorical data are expressed as frequencies and percentages. Comparisons among the three groups were conducted using one-way ANOVA for normally distributed continuous variables, and the Kruskal–Wallis test for non-normally distributed variables. For categorical variables, either the chi-square test or Fisher's exact test was applied, as appropriate. To minimize confounding due to baseline differences among groups, stabilized inverse probability of treatment weighting (IPTW) was applied. Propensity scores were

estimated using a multinomial logistic regression model based on baseline covariates, including age, gender, times of PTA, comorbidities (hypertension, diabetes mellitus, dyslipidemia, coronary artery disease, chronic kidney disease, end stage renal disease), and medications (statin, beta blocker, etc.). Stabilized weights were calculated as the marginal probability of group assignment divided by the propensity score for each individual. These weights were used to construct a pseudopopulation in which the distribution of baseline covariates was balanced across the three groups. After applying IPTW, covariate balance among the three groups was assessed using one-way ANOVA or Kruskal–Wallis tests for continuous variables and chi-square or Fisher's exact tests for categorical variables. These comparisons ensured that the weighted sample achieved adequate balance in baseline characteristics.

Cox proportional hazards models were used to estimate hazard ratios and 95% confidence intervals for outcomes across the groups. Both IPTW-weighted and unweighted models were constructed. In the unweighted models, multivariable adjustment for the aforementioned baseline covariates was performed. The proportional hazards assumption was verified prior to model fitting. A two-sided $p$-value $< 0.05$ was considered statistically significant. All statistical analyses were conducted using SAS version 9.4 (SAS Institute, Cary, NC, USA).

## Results

### Study population

A total of 17,993 patients diagnosed with LEAD and treated with PTA were retrospectively screened. Of these, 7,760 patients were excluded due to the occurrence of MACE within one year or MALE on the index date. After applying additional exclusion criteria, a final cohort of 5,300 patients was included in the analysis (mean age: 71 ± 12 years; 61% male). Among these patients, 1,786 (34%) received aspirin or clopidogrel monotherapy, 844 (16%) received cilostazol monotherapy, and 2,670 (50%) were treated with cilostazol in combination with aspirin or clopidogrel (Fig 1). The most notable baseline difference was the prevalence of coronary artery disease (CAD), which was significantly lower in the cilostazol group compared to the others (29% vs. 52%, $p < 0.01$, Table 1). This finding is clinically reasonable, as current guidelines recommend aspirin or clopidogrel for patients with CAD, making them less likely to be prescribed cilostazol monotherapy. In addition, despite their now-recognized therapeutic potential in the management of LEAD, newer pharmacological

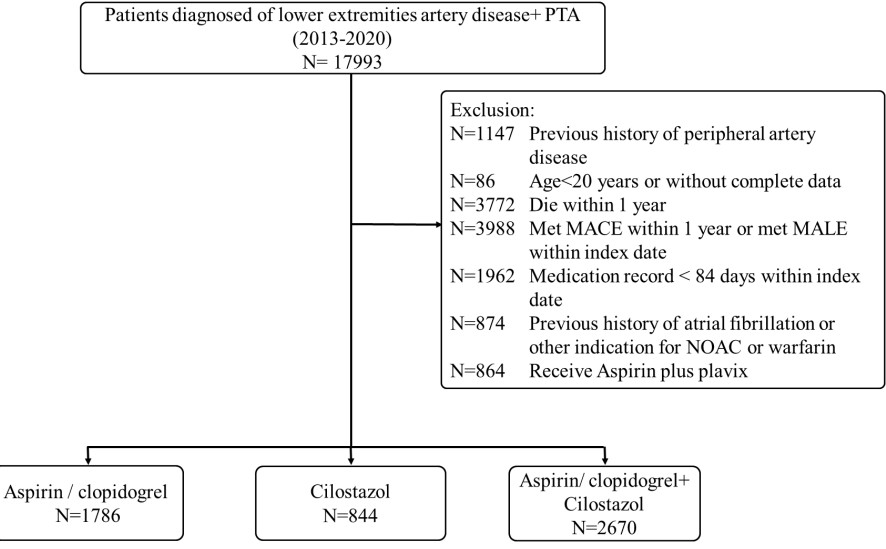

**Fig 1. Flow diagram of the study cohort.** (MACE: major adverse cardiovascular events; MALE: major adverse limb events; N: number of participants; NOAC: Novel oral anticoagulants; PTA: percutaneous transluminal angioplasty).

Table 1. The baseline characteristics of LEAD patients treated with before and after inverse probability of treatment weighting (IPTW).

| | Total | Inverse probability of treatment weighting | | | | | | | |
| | | Before | | | | After | | | |
| | | Aspirin or Plavix users | Cilostazol users | Aspirin+Cilostazol or Plavix+Cilostazol | P | Aspirin or Plavix users | Cilostazol users | Aspirin+Cilostazol or Plavix+Cilostazol | P |
| | N=5300 | N=1786 | N=844 | N=2670 | | N=pseudo data | N=pseudo data | N=pseudo data | |
|---|---|---|---|---|---|---|---|---|---|
| Age (mean±SD) | 71.21±11.61 | 70.78±11.43 | 72.21±12.30 | 71.19±11.49 | 0.34 | 71.13±11.51 | 71.00±12.01 | 71.23±11.51 | 0.39 |
| Gender (Male) | 3251 (61.34) | 1136 (63.61) | 471 (55.81) | 1644 (61.57) | | 61.47 | 62.12 | 61.42 | 0.94 |
| Times of PTA | | | | | <0.01 | | | | 0.89 |
| 1 | 4443 (83.83) | 1554 (87.01) | 729 (86.37) | 2160 (80.90) | | 83.72 | 84.53 | 83.79 | |
| 2 | 746 (14.08) | 207 (11.59) | 100 (11.85) | 439 (16.44) | | 14.15 | 12.95 | 14.18 | |
| ≥3 | 111 (2.09) | 25 (1.40) | 15 (1.78) | 71 (2.66) | | 2.13 | 2.51 | 2.03 | |
| Comorbidity | | | | | | | | | |
| Hypertension | 4446 (83.89) | 1518 (84.99) | 685 (81.16) | 2243 (84.01) | 0.04 | 83.88 | 84.10 | 83.82 | 0.98 |
| Diabetes mellitus | 3791 (71.53) | 1278 (71.56) | 583 (69.08) | 1930 (72.28) | 0.20 | 71.25 | 71.10 | 71.92 | 0.85 |
| Dyslipidemia | 2270 (42.83) | 776 (43.45) | 322 (38.15) | 1172 (43.90) | 0.01 | 43.03 | 42.83 | 42.70 | 0.98 |
| Coronary artery disease | 2407 (45.42) | 935 (52.35) | 250 (29.62) | 1222 (45.77) | <0.01 | 45.44 | 45.14 | 45.41 | 0.99 |
| Chronic kidney disease | 2439 (46.02) | 797 (44.62) | 400 (47.39) | 1242 (46.52) | 0.32 | 46.50 | 45.12 | 46.10 | 0.83 |
| End stage renal disease | 56 (1.06) | 19 (1.06) | 14 (1.66) | 23 (0.86) | 0.14 | 1.11 | 1.14 | 1.03 | 0.95 |
| Medications | | | | | | | | | |
| *Rivaroxaban 2.5 mg | 11 (0.21) | | | | – | – | | | – |
| Statin | 2728 (51.47) | 971 (54.37) | 333 (39.45) | 1424 (53.33) | <0.01 | 51.32 | 51.11 | 51.40 | 0.99 |
| Calcium channel blocker | 1415 (26.70) | 483 (27.04) | 222 (26.3) | 710 (26.59) | 0.91 | 27.43 | 26.62 | 26.45 | 0.77 |
| Beta blocker | 2287 (43.15) | 752 (42.11) | 312 (36.97) | 1223 (45.81) | <0.01 | 43.25 | 43.66 | 43.20 | 0.98 |
| ACEi/ARB | 2677 (50.51) | 995 (55.71) | 368 (43.60) | 1314 (49.21) | <0.01 | 50.65 | 50.46 | 50.46 | 0.99 |
| SGLT2 inhibitor | 187 (3.53) | 82 (4.59) | 18 (2.13) | 87 (3.26) | <0.01 | 3.60 | 3.41 | 3.58 | 0.98 |
| GLP-1RA | 69 (1.30) | 21 (1.18) | 11 (1.30) | 37 (1.39) | 0.83 | 1.12 | 1.61 | 1.39 | 0.60 |

Abbreviations: ACEi: Angiotensin-Converting Enzyme Inhibitor; ARB: Angiotensin II Receptor Blocker; GLP-1RA: Glucagon-Like Peptide-1 Receptor Agonist; PTA: Percutaneous Transluminal Angioplasty; SGLT2 Inhibitor: Sodium-Glucose Cotransporter 2 Inhibitor.

* Because the case number of thrombocytopenia was too small in one of the groups, the distribution of case number between groups was not allowed to be presented by the Taiwan Ministry of Health and Welfare.

agents such as rivaroxaban 2.5 mg, sodium-glucose cotransporter 2 (SGLT2) inhibitors, and glucagon-like peptide-1 (GLP-1) receptor agonists were prescribed in only 1–3% of patients in this real-world dataset. This likely reflects the treatment patterns of the earlier period during which the data were collected, prior to the widespread adoption of these agents based on more recent clinical evidence.

To account for differences in baseline characteristics—such as sex, number of PTA procedures, comorbidities, and concomitant medications—we applied IPTW. After adjustment, there were no significant differences among the three treatment groups (Table 1).

## Clinical outcomes

The mean follow-up time was 2.68 years. During this period, the overall incidence rates of MACE, MALE, and composite bleeding were 16.3%, 22.9%, and 13.8%, respectively. When comparing cilostazol monotherapy with aspirin or clopidogrel monotherapy, there were no statistically significant differences in the incidence of MACE (adjusted hazard ratio (aHR),

0.84; 95% CI, 0.68–1.03; $p = 0.09$) or MALE (aHR, 0.84; 95% CI, 0.70–1.01; $p = 0.06$, Table 2. Among the secondary end-points, cilostazol monotherapy was associated with a significantly lower risk of repeat PTA (aHR, 0.80; 95% CI, 0.66–0.98; $p = 0.03$). While there were no significant differences in the overall incidence of composite bleeding across groups, gastrointestinal bleeding appeared numerically lower in the cilostazol monotherapy group (aHR, 0.76; 95% CI, 0.57–1.01; $p = 0.05$). These findings suggest that cilostazol monotherapy may offer comparable clinical outcomes relative to aspirin or clopidogrel monotherapy in routine clinical practice.

In comparisons between aspirin or clopidogrel monotherapy and combination therapy with cilostazol, there were no statistically significant differences in the incidence of MACE (aHR, 0.87; 95% CI, 0.75–1.00; $p = 0.06$), MALE (aHR, 1.11; 95% CI, 0.97–1.26; $p = 0.12$), or composite bleeding (aHR, 0.94; 95% CI, 0.81–1.11; $p = 0.26$). The confidence interval for MACE approached the threshold for statistical significance.

**Table 2. Outcomes of LEAD patients received cilostazol and/or antiplatelet.**

| | Total (N = 5300) | | Aspirin or Plavix users. (ref.) (N = 1786) | | Cilostazol users (N = 844) | | Aspirin+Cilostazol or Plavix+Cilostazol (N = 2670) | | | *Non IPTW | | IPTW | |
|---|---|---|---|---|---|---|---|---|---|---|---|---|---|
| | | | | | | | | | | Adjusted HR (95% CI) | p value | HR (95% CI) | p value |
| Death | 1814 | (34.23) | 602 | (33.71) | 312 | (36.97) | 900 | (33.71) | C | 0.96 (0.84-1.10) | 0.56 | 0.88 (0.76-1.01) | 0.07 |
| | | | | | | | | | A+P | 0.91 (0.82-1.01) | 0.08 | 0.90 (0.81-0.99) | 0.04 |
| MACEs | 863 | (16.28) | 302 | (16.91) | 138 | (16.35) | 423 | (15.84) | C | 0.90 (0.73-1.10) | 0.30 | 0.84 (0.68-1.03) | 0.09 |
| | | | | | | | | | A+P | 0.86 (0.74-1.00) | 0.05 | 0.87 (0.75-1.00) | 0.06 |
| Cardiovascular death | 377 | (7.11) | 127 | (7.11) | 68 | (8.06) | 182 | (6.82) | C | 1.00 (0.74-1.35) | 0.98 | 0.89 (0.65-1.20) | 0.43 |
| | | | | | | | | | A+P | 0.87 (0.70-1.10) | 0.25 | 0.87 (0.70-1.09) | 0.22 |
| Non-fatal MI | 382 | (7.21) | 141 | (7.89) | 55 | (6.52) | 186 | (6.97) | C | 0.83 (0.60-1.14) | 0.25 | 0.80 (0.58-1.09) | 0.15 |
| | | | | | | | | | A+P | 0.82 (0.66-1.03) | 0.08 | 0.85 (0.68-1.06) | 0.14 |
| Ischemic stroke/TIA | 275 | (5.19) | 104 | (5.82) | 38 | (4.50) | 133 | (4.98) | C | 0.72 (0.49-1.05) | 0.09 | 0.74 (0.51-1.06) | 0.10 |
| | | | | | | | | | A+P | 0.81 (0.62-1.04) | 0.10 | 0.78 (0.60-1.00) | 0.05 |
| MALEs | 1213 | (22.89) | 383 | (21.44) | 163 | (19.31) | 667 | (24.98) | C | 0.84 (0.70-1.02) | 0.07 | 0.84 (0.70-1.01) | 0.06 |
| | | | | | | | | | A+P | 1.10 (0.97-1.25) | 0.13 | 1.08 (0.96-1.23) | 0.21 |
| Angioplasty for LEAD | 1087 | (20.51) | 342 | (19.15) | 136 | (16.11) | 609 | (22.81) | C | 0.80 (0.65-0.98) | 0.03 | 0.80 (0.66-0.98) | 0.03 |
| | | | | | | | | | A+P | 1.14 (0.99-1.30) | 0.06 | 1.11 (0.97-1.26) | 0.12 |
| Amputation | 325 | (6.13) | 108 | (6.05) | 53 | (6.28) | 164 | (6.14) | C | 0.95 (0.68-1.32) | 0.75 | 0.89 (0.64-1.25) | 0.50 |
| | | | | | | | | | A+P | 0.95 (0.74-1.21) | 0.66 | 0.96 (0.75-1.21) | 0.71 |
| Composite bleeding | 732 | (13.81) | 240 | (13.44) | 115 | (13.63) | 377 | (14.12) | C | 0.90 (0.72-1.13) | 0.36 | 0.88 (0.71-1.10) | 0.26 |
| | | | | | | | | | A+P | 0.96 (0.81-1.13) | 0.61 | 0.94 (0.81-1.11) | 0.48 |
| Hemorrhagic stroke | 45 | (0.85) | 17 | (0.95) | 6 | (0.71) | 22 | (0.82) | C | 0.66 (0.25-1.69) | 0.38 | 0.60 (0.24-1.53) | 0.28 |
| | | | | | | | | | A+P | 0.83 (0.44-1.57) | 0.57 | 0.77 (0.42-1.42) | 0.40 |
| Gastrointestinal bleeding | 470 | (8.87) | 160 | (8.96) | 65 | (7.7) | 245 | (9.18) | C | 0.76 (0.57-1.02) | 0.07 | 0.76 (0.57-1.01) | 0.05 |
| | | | | | | | | | A+P | 0.93 (0.76-1.14) | 0.47 | 0.91 (0.75-1.10) | 0.33 |
| Other site bleeding | 331 | (6.25) | 101 | (5.66) | 62 | (7.35) | 168 | (6.29) | C | 1.16 (0.84-1.61) | 0.36 | 1.12 (0.81-1.53) | 0.50 |
| | | | | | | | | | A+P | 1.02 (0.79-1.31) | 0.89 | 1.00 (0.79-1.28) | 0.99 |

Abbreviations: LEAD: CI, confidence interval; HR, hazard ratio; Lower Extremity Arterial Disease; MACE: Major Adverse Cardiovascular Events; MALE: Major Adverse Limb Events; MI: Myocardial Infarction; TIA: Transient Ischemic Attack.

* This model represents the pre-IPTW analysis, with adjustment for age, gender, times of PTA, comorbidity (hypertension, dyslipidemia, coronary artery disease), and medications (statin, beta blocker, ACEi/ARB, SGLT2 inhibitor).

## Subgroup analyses

Subgroup analyses were performed based on key baseline characteristics, including age, sex, times of PTA, diabetes, hypertension, dyslipidemia, coronary artery disease, chronic kidney disease, and end stage renal disease (Fig 2 and Supplemental Fig 2). Overall, the treatment effects were generally consistent across subgroups. The direction and magnitude of treatment effects remained similar in most strata, suggesting broadly consistent results across the study population.

## Discussion

In this real-world cohort of stable LEAD patients who underwent PTA, cilostazol therapy was associated with clinical outcomes comparable to those observed with aspirin or clopidogrel therapy. No statistically significant differences were noted in the incidence of MACE, MALE, or composite bleeding. In secondary analyses, cilostazol was associated with a significantly lower rate of repeat angioplasty, suggesting a potential benefit in reducing restenosis or recurrent limb ischemia.

LEAD remains a condition with a notably high mortality rate, surpassing that CAD and stroke. [12,13] A large cohort study reported event rates for all-cause mortality and MACE in LEAD patients to be approximately 113 and 71 per 1,000 person-years, respectively. In our cohort of 17,993 patients undergoing PTA—likely representing a population with more advanced disease—we observed a 21% one-year mortality rate and a 22% incidence of MACE or MALE. These strikingly high complication rates underscore the critical need for comprehensive care strategies that extend beyond revascularization alone. Early detection and the implementation of guideline-directed medical therapy are crucial cornerstones in managing LEAD patients. These findings further emphasize the importance of evaluating the potential benefits of individual pharmacologic therapies in improving outcomes in this high-risk population.

Cilostazol has been shown to exert multiple pleiotropic effects that may offer meaningful clinical benefits to patients with LEAD. Beyond its primary pharmacologic actions, inhibition of platelet aggregation and induction of vasodilation, cilostazol has demonstrated the ability to protect against apoptotic cell death, [14] anti-inflammation, [15] promote angiogenesis, and enhance endothelial function [6–8]. Clinically, these biological effects translate into improved patient

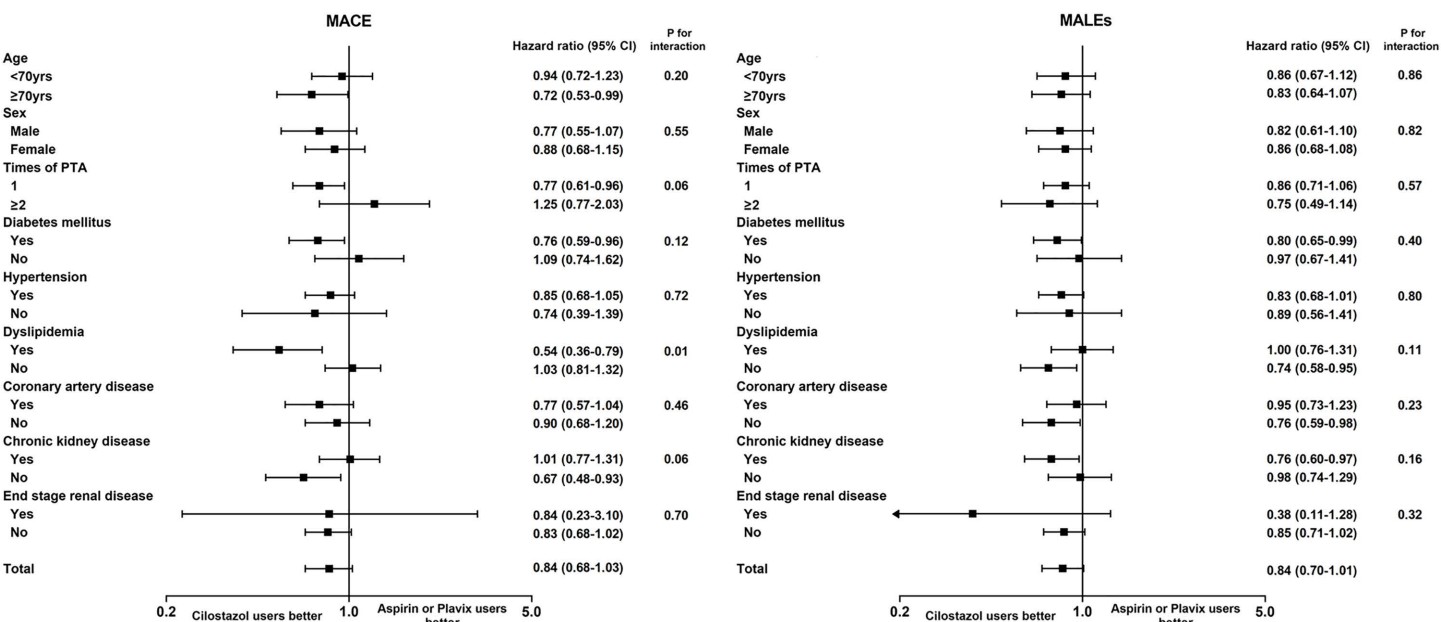

**Fig 2. Subgroup analysis of major adverse cardiovascular events and major adverse limb events between cilostazol users and aspirin or Plavix users.**

outcomes, including reduction in claudication symptoms, [16] increase walking distance, [17] and a lower risk of restenosis, re-occlusion, target lesion revascularization, [18] and even amputation. [19] Our findings complement and expand upon previous evidence by showing, in a real-world cohort, that cilostazol monotherapy was associated with a significantly lower rate of repeat PTA procedures compared to aspirin or clopidogrel monotherapy, without increasing the risk of MACE or bleeding. Although real-world data are inherently prone to selection bias, it is worth noting that cilostazol is often prescribed to patients with more pronounced symptoms. The observed reduction in MALE events may therefore reflect cilostazol's clinical effectiveness. Conversely, this same bias may partially explain why combination therapy with cilostazol and aspirin or clopidogrel did not show a statistically significant benefit over single-agent therapy, as those patients might have had more advanced disease at baseline.

In addition to its benefits in reducing MALE, cilostazol has also been evaluated for its potential role in lowering the risk of MACE. [20] A randomized controlled trial in high-risk cardiovascular patients reported a hazard ratio of 0.67 for MACE in the cilostazol group, suggesting a possible protective effect, particularly in subgroups with diabetes or prior percutaneous coronary intervention. [10] Similarly, a meta-analysis of 15 trials demonstrated a significant reduction in MACE risk (RR = 0.67; 95% CI, 0.56–0.81), although no difference in all-cause mortality was observed. [9] In hemodialysis patients with LEAD, cilostazol was associated with improved 10-year MACE-free survival (HR 0.57; 95% CI, 0.41–0.79). [21] In our real-world dataset, cilostazol monotherapy showed no significant difference in MACE risk compared to aspirin or clopidogrel. However, dual therapy combining cilostazol with either aspirin or clopidogrel yielded a hazard ratio for MACE (HR 0.87; 95% CI, 0.75–1.00) and stroke (HR 0.78; 95% CI, 0.60–1.00) that were numerically lower, though these did not meet the threshold for statistical significance. Subgroup analyses revealed generally consistent effects across various patient characteristics, further supporting the potential cardiovascular benefits of cilostazol in selected populations.

Several limitations should be considered when interpreting our findings. First, although we employed robust statistical adjustments using IPTW, the potential for residual confounding due to unmeasured variables—such as Rutherford classification, lesion severity, ankle-brachial index, degree of calcification, and lifestyle factors—cannot be entirely excluded. Second, as with all observational studies, the possibility of other uncontrolled sources of bias and confounding remains, warranting cautious interpretation of the results. Third, the data reflect clinical practice patterns between 2013 and 2020, a period when the adoption of newer cardiometabolic therapies—such as SGLT2 inhibitors, GLP-1 receptor agonists, and low-dose rivaroxaban (2.5 mg)—was limited. Therefore, the generalizability of our findings to more contemporary treatment paradigms may be constrained. Fourth, the use of a landmark design mitigated immortal time bias and strengthened internal validity, but may limit generalizability to patients with early post-PTA events.

In conclusion, this nationwide real-world analysis suggests that cilostazol monotherapy provides clinical outcomes and safety comparable to conventional antiplatelet therapy in stable LEAD patients following PTA. In addition, cilostazol was associated with a reduced need for repeat revascularization. These findings support its consideration as a viable therapeutic option in routine practice. Further randomized trials or prospective studies are warranted to confirm these results and clarify cilostazol's role within the broader context of modern LEAD management.

## Supporting information

**Supplemental Fig 1. Schematic diagram of the data collection timeline. (LEAD: lower extremities artery disease; MACE: major adverse cardiovascular events; MALE: major adverse limb events; N: number of participants; NOAC: Novel oral anticoagulants; PTA: percutaneous transluminal angioplasty).**
(TIF)

**Supplemental Fig 2. Subgroup analysis of major adverse cardiovascular events and major adverse limb events between cilostazol plus aspirin or Plavix users and aspirin or Plavix users.**
(TIF)

**S1 Table. Supplemental table 1. ICD-9 and ICD-10 codes.**
(DOCX)

## Author contributions

**Conceptualization:** Hsien-Yuan Chang, Po-Wei Chen, Ting-Hsing Chao, Yi-Heng Li.

**Data curation:** Hsien-Yuan Chang, Hui-Wen Lin, Sheng-Hsiang Lin, Yi-Heng Li.

**Formal analysis:** Hui-Wen Lin, Sheng-Hsiang Lin.

**Funding acquisition:** Ting-Hsing Chao.

**Investigation:** Hsien-Yuan Chang.

**Methodology:** Hsien-Yuan Chang, Sheng-Hsiang Lin.

**Project administration:** Hsien-Yuan Chang.

**Supervision:** Hsien-Yuan Chang, Sheng-Hsiang Lin, Ting-Hsing Chao, Yi-Heng Li.

**Validation:** Po-Wei Chen, Sheng-Hsiang Lin, Yi-Heng Li.

**Visualization:** Yi-Heng Li.

**Writing – original draft:** Hsien-Yuan Chang.

**Writing – review & editing:** Po-Wei Chen, Ting-Hsing Chao, Yi-Heng Li.

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
