## [Decision Letter · Decision Letter 0]

27 Jun 2025

Dear Dr. Li,

Thank you for submitting your manuscript to PLOS ONE. After careful consideration, we feel that it has merit but does not fully meet PLOS ONE’s publication criteria as it currently stands. Therefore, we invite you to submit a revised version of the manuscript that addresses the points raised during the review process.

We look forward to receiving your revised manuscript.

Kind regards,

Timir Paul

Academic Editor

PLOS ONE

Journal Requirements: 

 [This work was supported by a grant from the Chung Shan Medical University Hospital (CSH-2-25-D-002, Ting-Hsing Chao).]. 

5. We notice that your supplementary table is included in the manuscript file. Please remove them and upload them with the file type 'Supporting Information'. Please ensure that each Supporting Information file has a legend listed in the manuscript after the references list.

Reviewers' comments:

Reviewer's Responses to Questions

**Comments to the Author**

1. Is the manuscript technically sound, and do the data support the conclusions?

Reviewer #1: Yes

Reviewer #2: Yes

Reviewer #3: Yes

2. Has the statistical analysis been performed appropriately and rigorously?

Reviewer #1: Yes

Reviewer #2: I Don't Know

Reviewer #3: I Don't Know

3. Have the authors made all data underlying the findings in their manuscript fully available?

Reviewer #1: Yes

Reviewer #2: Yes

Reviewer #3: Yes

4. Is the manuscript presented in an intelligible fashion and written in standard English?

Reviewer #1: Yes

Reviewer #2: Yes

Reviewer #3: Yes

Reviewer #1: Chang et al. conducted a retrospective cohort study using Taiwan’s National Health Insurance Research Database (NHIRD) from 2012 to 2022. The study was designed to determine whether cilostazol, with or without standard antiplatelet therapy, improves long-term outcomes for patients with lower extremity arterial disease (LEAD) following angioplasty. The authors identified 5,300 stable patients—those who remained event-free for one year post-angioplasty—and applied stabilized inverse probability of treatment weighting (IPTW) and Cox regression to compare cilostazol monotherapy against aspirin/clopidogrel (and combination therapy).

The main findings revealed no statistically significant differences in major adverse cardiovascular events (MACE), major adverse limb events (MALE), or bleeding between the groups (adjusted hazard ratios ~0.84–0.88, p>0.05). However, there was a lower rate of repeat angioplasty in the cilostazol group (hazard ratio ~0.80, p=0.03). The authors concluded that cilostazol provides similar “prognostic benefits and safety” compared to standard therapy, supporting its use in the long-term management of LEAD. This topic is particularly relevant, given the existing gap between clinical trial evidence and guideline recommendations for cilostazol in LEAD.

Reviewer #2: General Comments

This study significantly contributes to understanding cilostazol's long-term effects in LEAD patients post-PTA in a real-world setting. Its strengths lie in using Taiwan's National Health Insurance Research Database (NHIRD), providing a comprehensive and representative view of clinical practice over ten years. The large sample size of 5,300 stable LEAD patients enhances statistical power and allows for robust analysis. The robust statistical methodology, including stabilized Inverse Probability of Treatment Weighting (IPTW) for confounder adjustment and a landmark design to mitigate immortal time bias, significantly strengthens internal validity.

Despite these strengths, inherent limitations exist. As an observational study, there is potential for residual confounding from unmeasured variables (e.g., Rutherford classification, lesion severity, lifestyle factors) not captured in the administrative database. The generalizability is constrained by its reliance on an Asian cohort and a data collection timeframe (2012-2022) that predates widespread adoption of newer cardiometabolic therapies, limiting applicability to contemporary clinical settings. Finally, the interpretation of statistically non-significant findings (e.g., p-values near 0.05) as "potential clinical relevance" requires more cautious phrasing, as it may over-interpret the evidence

Specific Comments

3.1. Title and Abstract

The title is precise, and the abstract is well-structured, summarizing key aspects and findings.

3.2. Introduction

The introduction effectively sets the clinical context, highlights the research gap, and clearly states the study's objective.

3.3. Methods

3.3.1. Study Design and Data Source

A retrospective cohort design utilized Taiwan's NHIRD (2012-2022). The 10-year period allows for long-term outcome evaluation, but evolving medical guidelines may limit contemporary generalizability.

3.3.2. Study Population and Inclusion/Exclusion Criteria

Adult LEAD patients post-PTA were included, identified by ICD codes. A landmark design (6-12 months post-PTA) mitigated immortal time bias. Exclusion of patients with early MACE/MALE focused on a stable cohort, improving internal validity for this subgroup but potentially limiting broader generalizability.

3.3.3. Clinical Outcomes

Primary endpoints were MACE and MALE; secondary endpoints included individual components and a composite bleeding safety outcome, all clearly defined.

3.3.4. Statistical Analysis

Stabilized IPTW successfully adjusted for baseline confounders, achieving covariate balance. Cox proportional hazards models were used, with the proportional hazards assumption verified. Potential residual confounding from unmeasured variables is acknowledged.

3.4. Results

Cilostazol monotherapy significantly reduced repeat PTA (aHR 0.80, p=0.03). A near-significant reduction in GI bleeding was also observed (aHR 0.76, p=0.05). The interpretation of p-values near 0.05 as "near-significant" or "possible directional benefit" is an over-interpretation and should be rephrased. Subgroup analyses generally showed consistent effects.

3.5. Discussion

Cilostazol therapy showed comparable outcomes and safety to conventional antiplatelet therapy, with a significant reduction in repeat angioplasty. High MACE/MALE rates in LEAD patients underscore the need for comprehensive care. Cilostazol's pleiotropic effects are discussed as potential mechanisms.

Limitations are transparently acknowledged: residual confounding from unmeasured variables, general observational study biases, and the data timeframe (2013-2020) limiting generalizability to contemporary practice due to limited adoption of newer therapies. The authors explicitly limit generalizability to Asian populations. Further studies are called for.

3.6. Conclusion

Cilostazol monotherapy provides comparable clinical outcomes and safety to conventional antiplatelet therapy in stable LEAD patients post-PTA, and was associated with reduced repeat revascularization. The phrasing "signals of potential benefit" for non-significant trends should be rephrased more cautiously.

Reviewer #3: The authors aim to compare cilostazol with other antiplatelet therapies for MACE, MALE, composite bleeding outcomes in patients with LEAD undergoing endovascular revascularization. The findings in the study add to current available literature. The limitations of the study are noted.

**Do you want your identity to be public for this peer review?** For information about this choice, including consent withdrawal, please see our Privacy Policy

Reviewer #1: **Yes: ** Roshan Bista

Reviewer #2: No

Reviewer #3: No

---

## [Author Response · Author response to Decision Letter 1]

24 Jul 2025

Responding letter

Reviewer #1:

Chang et al. conducted a retrospective cohort study using Taiwan’s National Health Insurance Research Database (NHIRD) from 2012 to 2022. The study was designed to determine whether cilostazol, with or without standard antiplatelet therapy, improves long-term outcomes for patients with lower extremity arterial disease (LEAD) following angioplasty. The authors identified 5,300 stable patients—those who remained event-free for one year post-angioplasty—and applied stabilized inverse probability of treatment weighting (IPTW) and Cox regression to compare cilostazol monotherapy against aspirin/clopidogrel (and combination therapy).

The main findings revealed no statistically significant differences in major adverse cardiovascular events (MACE), major adverse limb events (MALE), or bleeding between the groups (adjusted hazard ratios ~0.84–0.88, p>0.05). However, there was a lower rate of repeat angioplasty in the cilostazol group (hazard ratio ~0.80, p=0.03). The authors concluded that cilostazol provides similar “prognostic benefits and safety” compared to standard therapy, supporting its use in the long-term management of LEAD. This topic is particularly relevant, given the existing gap between clinical trial evidence and guideline recommendations for cilostazol in LEAD.

Reply: We thank the reviewer for the clear and accurate summary of our study. We appreciate the recognition of our efforts to evaluate cilostazol’s real-world effectiveness in the long-term management of LEAD. No changes were made to the abstract or main text in response to this comment, as the summary reflects the study design and findings accurately.

Reviewer: 2

General Comments

This study significantly contributes to understanding cilostazol's long-term effects in LEAD patients post-PTA in a real-world setting. Its strengths lie in using Taiwan's National Health Insurance Research Database (NHIRD), providing a comprehensive and representative view of clinical practice over ten years. The large sample size of 5,300 stable LEAD patients enhances statistical power and allows for robust analysis. The robust statistical methodology, including stabilized Inverse Probability of Treatment Weighting (IPTW) for confounder adjustment and a landmark design to mitigate immortal time bias, significantly strengthens internal validity.

Despite these strengths, inherent limitations exist. As an observational study, there is potential for residual confounding from unmeasured variables (e.g., Rutherford classification, lesion severity, lifestyle factors) not captured in the administrative database. The generalizability is constrained by its reliance on an Asian cohort and a data collection timeframe (2012-2022) that predates widespread adoption of newer cardiometabolic therapies, limiting applicability to contemporary clinical settings. Finally, the interpretation of statistically non-significant findings (e.g., p-values near 0.05) as "potential clinical relevance" requires more cautious phrasing, as it may over-interpret the evidence

Reply: Thank you for this valuable comment. We agree that interpertation of non-significant findings requires caution to avoid overstating. In response, we had revised the result section to “ In comparisons between aspirin or clopidogrel monotherapy and combination therapy with cilostazol, there were no statistically significant differences in the incidence of MACE (aHR, 0.87; 95% CI, 0.75–1.00; p = 0.06), MALE (aHR, 1.11; 95% CI, 0.97–1.26; p = 0.12), or composite bleeding (aHR, 0.94; 95% CI, 0.81–1.11; p = 0.26). The confidence interval for MACE approached the threshold for statistical significance.“. Similarly, in the discussion section, we had recives to “ However, dual therapy combining cilostazol with either aspirin or clopidogrel yielded a hazard ratio for MACE (HR 0.87; 95% CI, 0.75–1.00) and stroke (HR 0.78; 95% CI, 0.60–1.00) that were numerically lower, though these did not meet the threshold for statistical significance.“ These revisions reflect a more neutral and objective description of the data, and avoid interpretive language that may overstate the findings. We appreciate the reviewer’s guidance in strengthening the clarity and rigor of our reporting.

Specific Comments

3.1. Title and Abstract

The title is precise, and the abstract is well-structured, summarizing key aspects and findings.

Reply: Thank you for the positive feedback. No changes were made in response to this comment.

3.2. Introduction

The introduction effectively sets the clinical context, highlights the research gap, and clearly states the study's objective.

Reply: Thank you. We made no changes based on this comment.

3.3. Methods

3.3.1. Study Design and Data Source

A retrospective cohort design utilized Taiwan's NHIRD (2012-2022). The 10-year period allows for long-term outcome evaluation, but evolving medical guidelines may limit contemporary generalizability.

Reply: Thank you for highlighting this important point. We fully agree that evolving treatment guidelines and the introfuction of newer therapy may impact the generalizability of our finding. In the limitaiton, we had acknowlegdged this “ Third, the data reflect clinical practice patterns between 2013 and 2020, a period when the adoption of newer cardiometabolic therapies—such as SGLT2 inhibitors, GLP-1 receptor agonists, and low-dose rivaroxaban (2.5 mg)—was limited. Therefore, the generalizability of our findings to more contemporary treatment paradigms may be constrained.“ We believe this statemtn diredctly afresses the reviewre’s concern, and we are happy to revise or expand it further if needed.

3.3.2. Study Population and Inclusion/Exclusion Criteria

Adult LEAD patients post-PTA were included, identified by ICD codes. A landmark design (6-12 months post-PTA) mitigated immortal time bias. Exclusion of patients with early MACE/MALE focused on a stable cohort, improving internal validity for this subgroup but potentially limiting broader generalizability.

Reply: Thank you for your insightful comment. We fully agree that the use of a landmark design was appropriate to mitigate immortal time bias, given our focus on evaluating long-term pharmacologic effects. At the same time, we acknowledge that this approach may reduce generalizability to patients who experience early post-PTA events. To address this point, we had added the following sentence in limitatoin “Fourth, the use of a landmark design mitigated immortal time bias and strengthened internal validity, but may limit generalizability to patients with early post-PTA events.“

3.3.3. Clinical Outcomes

Primary endpoints were MACE and MALE; secondary endpoints included individual components and a composite bleeding safety outcome, all clearly defined.

Reply: Thank you. No changes were needed.

3.3.4. Statistical Analysis

Stabilized IPTW successfully adjusted for baseline confounders, achieving covariate balance. Cox proportional hazards models were used, with the proportional hazards assumption verified. Potential residual confounding from unmeasured variables is acknowledged.

Reply: Thank you for the observation. We agree and have already acknowledged the possibility of residual confounding from unmeasured variables in the limitations section.

3.4. Results

Cilostazol monotherapy significantly reduced repeat PTA (aHR 0.80, p=0.03). A near-significant reduction in GI bleeding was also observed (aHR 0.76, p=0.05). The interpretation of p-values near 0.05 as "near-significant" or "possible directional benefit" is an over-interpretation and should be rephrased. Subgroup analyses generally showed consistent effects.

Reply: Thank you for this helpful suggestion. We agree that p-values close to 0.05 should be interpreted with caution. In response, we had revised the sentence in the Results section to avoid the phrase “near-significant” and to use more neutral language. “While there were no significant differences in the overall incidence of composite bleeding across groups, gastrointestinal bleeding appeared numerically lower in the cilostazol monotherapy group (aHR, 0.76; 95% CI, 0.57–1.01; p = 0.05).“

3.5. Discussion

Cilostazol therapy showed comparable outcomes and safety to conventional antiplatelet therapy, with a significant reduction in repeat angioplasty. High MACE/MALE rates in LEAD patients underscore the need for comprehensive care. Cilostazol's pleiotropic effects are discussed as potential mechanisms.

Reply: We sincerely thank the reviewer for this clear and concise summary, which highlights the main findings and key discussion points of our study. As these aspects have already been thoroughly addressed in the manuscript, no further revisions were made.

Limitations are transparently acknowledged: residual confounding from unmeasured variables, general observational study biases, and the data timeframe (2013-2020) limiting generalizability to contemporary practice due to limited adoption of newer therapies. The authors explicitly limit generalizability to Asian populations. Further studies are called for.

Reply: We appreciate the reviewer’s acknowledgement of our transparent discussion on study limitations and the scope of generalizability. We concur that further research is warranted to extend these findings to broader populations and evolving treatment paradigms.

3.6. Conclusion

Cilostazol monotherapy provides comparable clinical outcomes and safety to conventional antiplatelet therapy in stable LEAD patients post-PTA, and was associated with reduced repeat revascularization. The phrasing "signals of potential benefit" for non-significant trends should be rephrased more cautiously.

Reply: We thank the reviewer for the insightful comment. We agree that given the subgroup analyses and potential selection bias, non-significant trends should be interpreted with caution. Accordingly, we have removed the phrase “signals of potential benefit” from the Conclusion section. The revised conclusion now states: “In addition, cilostazol was associated with a reduced need for repeat revascularization.” We believe this change makes the manuscript’s interpretation more neutral and scientifically appropriate.

Reviewer: 3

The authors aim to compare cilostazol with other antiplatelet therapies for MACE, MALE, composite bleeding outcomes in patients with LEAD undergoing endovascular revascularization. The findings in the study add to current available literature. The limitations of the study are noted.

Reply: We sincerely thank the reviewer for the positive evaluation and recognition of our study’s contribution to the current literature. We also appreciate the acknowledgment of the study’s limitations, which we have addressed in the Discussion section. No additional changes were required in response to this comment.

---

## [Editor Report · Decision Letter 1]

1 Aug 2025

Impact of Cilostazol on Clinical Outcomes in Lower Extremity Arterial Disease Patients After Angioplasty: A Real-World Analysis

PONE-D-25-23547R1

Dear Dr. Li,

We’re pleased to inform you that your manuscript has been judged scientifically suitable for publication and will be formally accepted for publication once it meets all outstanding technical requirements.

Kind regards,

Timir Paul

Academic Editor

PLOS ONE

---

## [Editor Report · Acceptance letter]

PONE-D-25-23547R1

PLOS ONE

Dear Dr. Li,

I'm pleased to inform you that your manuscript has been deemed suitable for publication in PLOS ONE. Congratulations! Your manuscript is now being handed over to our production team.

Kind regards,

on behalf of

Dr. Timir Paul

Academic Editor

PLOS ONE